# The 5:2 Diet Affects Markers of Insulin Secretion and Sensitivity in Subjects with and without Type 2 Diabetes—A Non-Randomized Controlled Trial

**DOI:** 10.3390/ijms25179731

**Published:** 2024-09-08

**Authors:** Neda Rajamand Ekberg, Anton Hellberg, Michaela Linn Sundqvist, Angelica Lindén Hirschberg, Sergiu-Bogdan Catrina, Kerstin Brismar

**Affiliations:** 1Department of Molecular Medicine and Surgery, Karolinska Institutet, 171 77 Stockholm, Sweden; neda.ekberg@ki.se (N.R.E.); sergiu.catrina@ki.se (S.-B.C.); kerstin.brismar@ki.se (K.B.); 2Centre for Diabetes, Academic Specialist Centre, 113 65 Stockholm, Sweden; 3Department of Endocrinology, Metabolism and Diabetes, Karolinska University Hospital, 171 64 Stockholm, Sweden; 4Department of Women’s and Children’s Health, Karolinska Institutet, 171 77 Stockholm, Sweden; angelica.hirschberg.linden@ki.se; 5Department of Physiology, Nutrition and Biomechanics, Åstrand Laboratory, The Swedish School of Sport and Health Sciences, 114 33 Stockholm, Sweden; michaela.sundqvist@gih.se; 6Department of Gynecology and Reproductive Medicine, Karolinska University Hospital, 171 76 Stockholm, Sweden

**Keywords:** intermittent fasting, 5:2 diet, type 2 diabetes, insulin sensitivity, IGFBP-1

## Abstract

This non-randomized controlled trial aimed to compare the effect of the 5:2 diet on insulin levels as a primary outcome and markers of insulin secretion (connecting peptide (C-peptide) and insulin-like growth factor binding protein-1 (IGFBP-1)) and sensitivity (Homeostatic Model Assessment for Insulin Resistance (HOMA-IR)), as well as body composition as secondary outcomes in overweight/obese individuals with and without type 2 diabetes (T2D). Ninety-seven participants (62% women), 35 with T2D and 62 BMI- and waist-matched controls without T2D, followed the 5:2 diet (two days per week of fasting) for six months with a 12-month follow-up. At six months, there was no loss to follow-up in the T2D group, whereas four controls discontinued this study. Overall, 82% attended the 12-month follow-up. After the intervention, insulin levels decreased in the control group and glucose decreased in the T2D group, while C-peptide, HOMA-IR, waist circumference, BMI, trunk, and total fat% decreased in both groups. Furthermore, low IGFBP-1, indicating hyperinsulinemia, improved in the T2D group. The changes in fasting glucose and waist measurement were significantly more improved in the T2D group than in the controls. Persistent positive effects were observed at the 12-month follow-up. The 5:2 diet for six months was feasible and efficient to reduce markers of insulin secretion and resistance and therefore holds promise as management of overweight/obesity in subjects with and without T2D.

## 1. Introduction

Type 2 diabetes (T2D) affects approximately 10.5% of the adult population globally [1]. The prevalence of overweight and obesity, major risk factors for T2D, is increasing and is recognized as a significant contributor to the global disease burden [2,3,4]. A high-calorie diet induces hyperinsulinemia, which stimulates the growth of the adipose tissue and thus leads to overweight and obesity. The association between obesity and T2D is partly attributed to visceral obesity and ectopic fat, which are linked to hyperinsulinemia and insulin resistance (IR). IR explains elevated blood glucose, high blood pressure, and dyslipidemia, comprising the metabolic syndrome, which elevates the risk of both cardiovascular disease (CVD) and T2D [5,6,7,8,9].

Weight loss interventions through calorie restriction (CR) are known to reduce insulin secretion and improve insulin sensitivity and CVD risk factors in obese individuals with and without T2D [10,11,12,13]. However, traditional CR often fails as a long-term intervention due to poor compliance [14]. Weight loss surgery is costly, associated with side effects, and provides limited benefits at the population level [15,16]. Pharmacotherapy with glucagon-like peptide-1 receptor agonists is effective during treatment but is also associated with high costs, side effects, and lacks long-term studies on its efficacy. While long-term energy-restrictive diets are challenging, less restrictive approaches integrated into a healthy lifestyle are preferable. Intermittent fasting (IF), alternating CR with non-fasting periods, has gained popularity in recent years as it is believed to offer similar benefits as traditional continuous CR but with improved compliance [17,18]. Previous short-term studies of IF (three months) have shown positive effects on clinical outcomes [19,20,21], although results regarding markers of insulin resistance and HbA1c (glycated hemoglobin) have been conflicting [22].

Circulating insulin is dependent on hepatic extraction and hepatic insulin resistance, and therefore fasting connecting peptide (C-peptide), a marker of insulin secretion [23], is important to analyze in T2D. Furthermore, glucose homeostasis is not only regulated by insulin but also by Insulin-like Growth Factor-I (IGF-I), which increases insulin sensitivity [24]. Insulin elevates free IGF-I levels by inhibiting at the transcriptional level the synthesis of insulin-like growth factor binding protein-1 (IGFBP-1) [25,26,27]. Thus, IGFBP-1 is a marker of insulin secretion. Low fasting IGFBP-1 levels indicate high insulin secretion [24,26], hepatic [28], and whole-body IR [29], and predict the development of T2D [30,31,32,33]. Homeostatic Model Assessment for Insulin Resistance (HOMA-IR) based on fasting insulin and glucose levels is an established measurement of insulin resistance [34].

This non-randomized controlled trial aimed to compare the effects of the 5:2 diet during six months, with a 12-month follow-up on the insulin levels as the primary outcome and markers of insulin secretion (C-peptide and IGFBP-1) and sensitivity (HOMA-IR) and body composition as secondary outcomes in overweight/obese men and women with and without T2D.

## 2. Results

### 2.1. Baseline Characteristics

Baseline characteristics of the two groups are displayed in Table 1. The groups were body mass index (BMI)- and waist-matched, but the participants with T2D were on average older than the controls. Age-adjusted fasting insulin and IGFBP-1 were comparable between groups, but participants with T2D had significantly higher levels of age-adjusted fasting glucose, C-peptide, HOMA-IR, and HbA1c.

### 2.2. Six Months of Intervention

In total, 93 out of 97 participants (96%) completed the intervention for six months (Figure 1). In the T2D group, there was no one lost to follow-up at six months (n = 35, 66% women), whereas in the control group (n = 62, 63% women), four participants discontinued the intervention (6.5%) for reasons listed in Figure 1.

After six months of intervention, the T2D group displayed a significant weight loss of 6.3%, and the corresponding weight loss in the control group was 5.0% (Table 2). Furthermore, there were significant reductions in waist circumference, body fat percentage, and trunk fat percentage in both groups (*p* < 0.001, respectively) without any significant differences between groups except for waist, which was significantly more reduced in the T2D group (Table 2).

Fasting glucose and HbA1c decreased significantly only in the T2D group (*p* < 0.001 and *p* = 0.004, respectively), while insulin levels decreased significantly only in the control group (*p* = 0.02). C-peptide decreased in both groups (T2D: *p* = 0.03, controls: *p* = 0.04) (Table 2). IGF-SD levels decreased significantly only in the controls (*p* = 0.001) and IGFBP-1 levels increased only in the T2D group (*p* = 0.04), while HOMA-IR decreased in both groups (T2D: *p* = 0.02, controls: *p* = 0.01). The changes in waist circumference, fasting glucose, and HbA1c were significantly greater in the T2D group compared to the control group (*p* = 0.02, *p* = 0.004, and *p* = 0.002, respectively), and the change in IGF-SD was also significantly different between groups (*p* < 0.001).

### 2.3. Compliance of Diet and Physical Activity

All participants continuously reported in their food diaries what menu they had chosen of the recommended dietary options on the fasting days during the intervention. Furthermore, self-reported physical activity was not significantly impacted by study participation.

### 2.4. Twelve Months of Follow-Up

There was a 12-month follow-up, six months after completed intervention and food intake ad libitum. In total, 82 participants (84%) came to the visit at 12-month follow-up. Four (three women) participants were lost to follow-up in the T2D group and seven (five women) in the control group (Figure 1). A total of 24 participants (30%) continued to use the 5:2 diet, 23 occasionally (28%), and 34 (42%) stopped using the 5:2 diet after the initial six months. One participant did not answer the question about continued use of the 5:2 diet.

At 12-month follow-up, there were small but significant increases in body weight in both groups compared with that after six-month intervention (T2D: *p* = 0.04, Controls: *p* = 0.002), and also an increase in body fat percentage in the T2D group (*p* = 0.02). However, there were still significant reductions in body weight, waist, body fat, and trunk fat compared to baseline in both groups (Table 3).

Fasting glucose was still reduced in both groups and insulin in the control group, while there was a tendency of decreased insulin levels in the T2D group (*p* = 0.06). C-peptide was no longer reduced in the two groups at 12-months compared to baseline (Table 3). However, HOMA-IR was still significantly decreased in both groups at the 12-month follow-up (Table 3, Figure 2). IGF-SD levels were significantly increased in the T2D group compared to baseline, but IGFBP-1 levels and HbA1c were not significantly changed. The HOMA-IR value was improved in T2D but not normalized at the 12-month visit, while the HOMA-IR was further improved to reach more normal values at 12 months in the controls.

## 3. Discussion

The 5:2 diet, followed for six months, effectively improved markers of insulin secretion (C-peptide in all and IGFBP-1 in T2D) and insulin sensitivity (HOMA-IR), as well as body weight and body composition in overweight/obese subjects with T2D and controls. Moreover, persistent positive effects were observed at the 12-month follow-up. This study is one of the few long-term studies on the 5:2 diet and the first to report the diet’s effect on several hormonal and metabolic risk markers, including IGF-I and IGFBP-1, in individuals with T2D and controls.

The diet was particularly beneficial for those with T2D, showing more improvements in fasting glucose and waist measurement than the controls, and with no dropout at six months. Although the controls were matched by BMI and waist circumference, the mean loss in waist was significantly more pronounced in the T2D subjects, with a tendency towards greater loss also in BMI compared to the controls. The mean body weight loss after the six-month 5:2 diet observed in the present study (6.2% in the T2D subjects and 5.0% in the controls) is in line with previous clinical trials on intermittent fasting [35,36,37].

The 5:2 diet, followed for six months, did not improve the insulin levels significantly among the subjects with T2D, but the decrease in fasting glucose, C-peptide, HOMA-IR, and HbA1c and increase in IGFBP-1 suggest beneficial effects of the 5:2 diet for glycemic control in T2D. Thus, after six months of intervention, the same insulin secretion was sufficient to reduce fasting glucose in T2D. C-peptide and insulin are secreted to v. porta in equimolar amounts from the beta cells when stimulated by high glucose. C-peptide escapes the hepatic extraction, while circulating insulin measured is the result of both insulin secretion and the insulin degradation in the liver [23]. The blunted decrease in insulin but not in C-peptide suggests that the reduced hepatic insulin extraction was not improved by the 5:2 diet in T2D in contrast to that in the controls. In the controls, the reduced insulin secretion was enough to keep fasting glucose within the normal range. Wang et al. found similar improvements in metabolic control in T2D when comparing CR and IF in a systematic review and meta-analysis [35]. The decrease in C-peptide levels and HOMA-IR in our controls is in line with the result of the 5:2 diet studied by Harvie et al. [38].

Moreover, persistent positive effects were observed at the 12-month follow-up. The majority of the participants reported that they had occasionally followed the 5:2 diet up to the 12-month visit. The weight loss was still significant in both groups, and so was the decrease in waist circumference and body and trunk fat percent. Furthermore, the insulin resistance measured by HOMA-IR was still significantly lower compared to baseline for both groups. The C-peptide levels, however, had increased to the baseline levels. The decreased glucose levels in both groups compared to baseline and reduced insulin in the control group, as well as the same tendency in the T2D group, suggest that the hepatic insulin extraction had increased. An increased insulin extraction suggests improved hepatic insulin sensitivity, which explains the improved fasting glucose levels in both groups.

The IGFBP-1 levels increased significantly in the T2D group after six months of intervention, which is possibly explained by reduced C-peptide/insulin secretion and improved hepatic insulin sensitivity due to reduced liver fat accumulation [28]. The decrease in IGF-SD observed in the controls in this study contrasts with a previous study by Harvie et al. reporting no change in IGF-I after three months of a 5:2 diet in overweight women [38]. The control subjects in our study were older and included both men and women, and the intervention was longer, which may explain the observed difference in the results. An increase in IGF-I and IGFBP-1 in subjects with T2D has been reported after weight loss following bariatric surgery [39]. The effect on IGF-I may be via the increased IGFBP-1, which stimulates GH secretion.

HbA1c is an important diabetes marker of both postprandial and fasting glucose over several weeks [40], while fasting glucose is a marker of hepatic gluconeogenesis regulated by hepatic insulin sensitivity. HbA1c was reduced after six-month intervention in the T2D subjects but not in the controls having low to normal HbA1c levels. At 12 months, HbA1c had increased in the T2D group, although not fully to the baseline level, and was no longer significantly decreased compared to baseline. This may be explained by the moderate increase in weight. However, the majority of the T2D subjects had in general well-controlled diabetes already before the start of the intervention.

A recent review of IF by Varady et al. [22] concluded that long-term, controlled studies of IF and the 5:2 diet are needed to further elucidate the efficacy and safety of these diets in different populations. Significant improvements in metabolic health and body composition were achieved in the present study with minimal support and costs (three visits in six months). Other studies with few visits showed less response, i.e., lower weight loss [41]. Our study demonstrated, in comparison to most studies on CR, a minimal dropout rate, and none with T2D discontinued the intervention despite no extra support between the visits. Furthermore, self-reported data on diet and physical activity support good compliance to the 5:2 diet, and no adverse effects or events were reported.

Some limitations must be acknowledged. The absence of a comparator dietary arm, such as daily CR, limits direct comparison with previous studies. However, the weight reduction observed in this study aligns with other IF interventions [19,42]. The aim of the present study was not to compare the response to the 5.2 diet with other diets but to compare the responses to the diet in subjects with T2D with BMI- and waist-matched subjects with normal fasting blood glucose and HbA1c. There was a significant age difference between the two groups, and therefore age adjustments were performed in the statistical analyses. Furthermore, ITT analyses were performed. Daily calorie intake and physical activity were not directly monitored but assessed through questionnaires and food diaries recorded on the fasting days. We did not measure beta-hydroxybutyrate or ketones during the fasting days, which could have supported a fasting state; only information regarding what menu the participants had chosen during the fasting days was collected. Another limitation was not to perform an insulin tolerance test or an oral glucose tolerance test. Instead, IGFBP-1 was used as a marker for hepatic and whole-body insulin sensitivity and IGF-I as a marker for insulin secretion.

Strengths of the study include its duration (six months), one of the longest interventions of the effect of the 5:2 diet in subjects with T2D compared to controls, and a 12-month follow-up. Most previous studies investigating the effect of the 5:2 diet had a shorter duration (3–12 weeks), and most of them included only obese participants without diabetes with some exceptions [19,36].

In summary, adhering to the 5:2 diet for six months was feasible with a low dropout rate, none among those with T2D, resulting in reduced insulin production with reduced C-peptide levels and improved whole-body and hepatic insulin sensitivity for both individuals with and without T2D. The effects of the diet were also still evident at the 12-month follow-up. This dietary intervention proved to be safe for individuals with T2D who were not on sulfonylureas or insulin, and it holds promise as a complementary approach in T2D management but also as an intervention for overweight/obesity individuals without T2D. Nonetheless, further investigations involving longer-term interventions with the 5:2 diet in larger cohorts of individuals with T2D are warranted to fully understand its effects on T2D. Participating in this study provided the subjects with T2D an additional tool to control their blood glucose and body weight. However, this study cannot tell whether this diet is better for those with T2D than any other diet with calorie restriction.

## 4. Materials and Methods

### 4.1. Study Design

This is a non-randomized controlled trial comparing the effect of the 5:2 diet on markers of insulin secretion and sensitivity in overweight/obese individuals with and without T2D. The intervention lasted for 6 months, and there was a 12-month follow-up. This study adhered to the regulations governing human studies in Sweden and received ethical approval from the Regional Ethical Review Board in Stockholm (Ethical approval number Dnr:2013/1618-31/3). It was also registered with ClinicalTrials.gov under identification number NCT02450097.

### 4.2. Eligibility Criteria

Inclusion criteria: Participants aged 18 years or older, with or without T2D, and BMI ranging from 25 to 37 kg/m^2^, with waist circumference exceeding 80 cm for women and 94 cm for men.

Exclusion criteria: BMI below 25 kg/m^2^, waist circumference less than or equal to 80 cm for women and 94 cm for men, treatment with insulin or sulfonylurea, conditions where weight reduction was undesired, chronic kidney disease (defined as an Estimated Glomerular Filtration Rate (eGFR) less than 30 mL/min), pregnancy or breastfeeding, active involvement in athletics, history of eating disorders, cancer diagnosis, or participation in another ongoing study.

### 4.3. Subjects

Subjects were enlisted via announcements, and inclusion screening was carried out using a digital questionnaire. All subjects gave informed consent before the enrollment in this study. Screening was conducted in cohorts of ten to twelve individuals. The aim was to recruit 40 subjects with T2D and 80 non-diabetic overweight/obese subjects matched for BMI and waist circumference. Since the inclusion took longer than planned, it was stopped when a total of 104 subjects (37 with T2D and 67 controls) had undergone screening. Of these 104 subjects, seven discontinued and became screening failures. One participant did not fulfill the study criteria, and six participants discontinued for other reasons such as pregnancy (n = 1), another serious diagnosis (n = 1), not being able to adhere to the diet (n = 2), and for unknown reasons (n = 2). Thus, 97 subjects were enrolled in the 6-month trial, 35 with T2D and 62 controls; among them, 39 (63%) of the controls and 23 (66%) of the subjects with T2D were women.

Treatment with sulfonylurea was discontinued in three subjects with T2D before their enrollment in this study. Among the 35 participants with T2D, 13 were solely treated with diet, 10 were on monotherapy with metformin, eight were treated with metformin in combination with a DPP4-inhibitor, two with metformin in combination with a GLP-1RA (semaglutide), one with metformin in combination with a Sodium/Glucose Cotransporter inhibitor (SGLT2-i), and one with metformin, SGLT2-i, and GLP-1RA at baseline.

### 4.4. Procedure

This study was conducted at the Department of Endocrinology, Metabolism, and Diabetes, Karolinska University Hospital, between 2013 and 2015 and performed in accordance with Good Clinical Practice (GCP) guidelines.

Participants visited the clinic before the intervention started (baseline) and after 6, 12, and 24 weeks (final visit). After the completion of the intervention, participants were invited to attend a 12-month follow-up visit with no dietary restrictions. Participants were only asked to keep track of their dietary habits and weigh themselves weekly. A physician examined each subject at the baseline visit. Thereafter, all visits were scheduled after non-fasting days when the participants were examined by a nurse. At baseline, 3 and 6 months, the following assessments were performed in the morning after an overnight fast: Blood pressure was measured twice on the left arm using an automatic reader (Boso-medicus control, Bosch, Germany) after a 10-minute rest in a seated position, with the lowest value recorded for analysis; blood samples were collected for measurement of metabolic variables; waist circumference was measured midway between the lower rib margin and the superior anterior iliac crest in a standing position; and body weight, total body fat, and trunk fat percentage were measured using a bioimpedance scale (Tanita body composition analyzer BC-418, Tokyo, Japan). BMI was calculated as weight in kilograms divided by the square of height in meters (kg/m^2^). At each visit, the participants were asked to bring completed questionnaires regarding physical activity, self-rated health, and quality of life, as well as food diaries for the fasting days, to the visits.

### 4.5. Dietary Intervention

Before the intervention started, a group meeting was held with 10–14 participants, one or two physicians, two nurses, and a dietitian. During this meeting, participants were informed about the study design, background information, and current Nordic Nutrition Recommendations, including the Mediterranean and Nordic diets. Study participants were instructed to follow the 5:2 diet, which involved two non-consecutive fasting days per week. On the remaining five days, participants were advised to follow a Mediterranean or Nordic diet ad libitum, based on the general information they received. The fasting days entailed consuming 500 kcal (2092 kilojoules) per day for women and 600 kcal (2510 kilojoules) per day for men, with a minimum intake of 20 g protein per day for both sexes. Calorie-free or very low-calorie drinks such as water, coffee, and tea were allowed without restriction. During the group meeting, participants were provided with recipes for 20 meals, each containing 100–400 kcal (419–1674 kilojoules) per meal, to choose from on fasting days. These meals were high in fiber and protein and low in fat, comprising vegetables, whole grain products, legumes, eggs, lean meat, and low-fat dairy products, with the aim of maximizing satiety, minimizing cooking time, and providing high nutritional value. To add variety, participants could choose from two or three meals per day from the menus, which included five breakfast options, ten main meals, and five snack options. Main meals included vegetarian, fish, red meat, and poultry dishes, while breakfast and snack options were based on whole grain products, dairy products, or eggs, all containing vegetables or fruits. Adherence to the recommended dietary strategy would result in a weekly reduction of ca. 20–25% of baseline energy intake. No specific advice was given regarding physical activity. To monitor compliance with the diet, participants were asked to report weekly on their food intake on fasting days in food diaries, including their meal choices. On fasting nights (around 10 pm), participants rated their feelings of hunger, satiety, and their ability to follow the diet using Visual Analog Scales (VAS).

### 4.6. Questionnaires

Physical activity (PA) levels were assessed using a validated questionnaire covering four distinct domains: work-related PA, transportation to and from work, leisure-time PA, and total PA [43,44].

Self-Rated Health (SRH) was assessed (but not reported here) using questions concerning participants’ overall perception of their health and their health compared to others of the same age using Cantril’s Ladder of Life, where respondents rated their current quality of life (QoL), QoL over the past year, and anticipated QoL one year from now on a 10-point Likert scale ranging from “Worst imaginable life” = 1 to “Best possible life” = 10 [45].

### 4.7. Laboratory Analysis

Fasting levels of insulin were measured by the clinical standard method electrochemiluminescence immunoassay (ECLIA) [Roche Cobas 8000 (e602)] at the Karolinska University Laboratory with a reference range of 2.0–25 mIU/L. Levels of insulin at fasting between 14–25 mIU/L indicate insulin resistance. Glucose levels were analyzed as routine at the Karolinska University Hospital laboratory. Fasting C-peptide was determined immunochemically with Roche Modular E by the Karolinska University Hospital laboratory with a reference range of 0.25–1.0 nmol/L. Serum IGF-I was determined by an in-house radioimmunoassay (RIA) after alcohol-acid extraction and cryoprecipitation [46]. Des (1–3) IGF-I was used as a radio-ligand. The detection limit was 6 μg/L, and intra- and inter-assay CVs were 4% and 11%, respectively. IGF-I levels were expressed as SD scores (IGF-SD) calculated from the regression of the IGF-I vales of healthy adult subjects aged 20–95 years [47]. This is necessary as a linear inverse correlation exists between logarithmically transformed IGF-I values and increasing age in healthy adults with no gender differences [47]. IGFBP-1 concentrations were determined by an in-house RIA [48]. The intra- and inter-assay CV were 3% and 10%, respectively. The detection limit was 3 μg/L. HOMA-IR was calculated using the formula (insulin mIU/L × glucose mmol/L/22.5) according to Matthews et al. [34]. A value ≥ 2.9 indicates insulin resistance. HbA1c levels were analyzed as routine at the Karolinska University Hospital laboratory using capillary electrophoresis. Normal reference values for individuals above 50 years are 31–46 mmol/mol.

### 4.8. Outcomes

The primary outcome of the intervention was to assess changes in insulin levels between baseline and 6 months of intervention. Secondary outcomes included changes in markers of insulin secretion (C-peptide, IGFBP-1 levels) and markers of insulin sensitivity (HOMA), as well as changes in body composition (waist circumference, total body fat, and trunk fat).

### 4.9. Statistics

All statistical analyses were performed using the RStudio statistical program (Version 1.4.1106). Tests for normality were assessed using the Shapiro–Wilk test, test of skew, analysis of histograms, and QQ plots. Independent sample’s *t*-test and Wilcoxon rank sum test, when appropriate, were used to determine baseline differences between the two study groups. When comparing the groups regarding the variables glucose, insulin, C-peptide, IGF-SD, IGFBP-1, HOMA, and HbA1c at baseline, a linear model was used with age as a covariate. Mixed Model ANOVA, with the factors Group, Time, and the interaction Group*Time, was used to analyze the differences within and between groups. These analyses are based on the procedure intention-to-treat (ITT). The results presented in Table 2 and Table 3 are adjusted for age as a covariate.

Log transformation was performed prior to Mixed Model ANOVA for variables that demonstrated positive skewness and are presented as antilog values in Table 2 and Table 3. Graphically, antilog values of HOMA-IR and insulin are presented with a geometric mean and 95% confidence interval. The variable age did not meet the assumption of equal variances, and the mean difference was calculated using Satterthwaite–Welch correction. In Table 1, anthropometric values are presented as mean ± SD. Non-normally distributed values distributed are presented as median and interquartile range (25th–75th). The results in Table 2 and Table 3 are presented as mean and 95% confidence interval (CI). A *p*-value < 0.05 was considered significant.

## Figures and Tables

**Figure 1 ijms-25-09731-f001:**
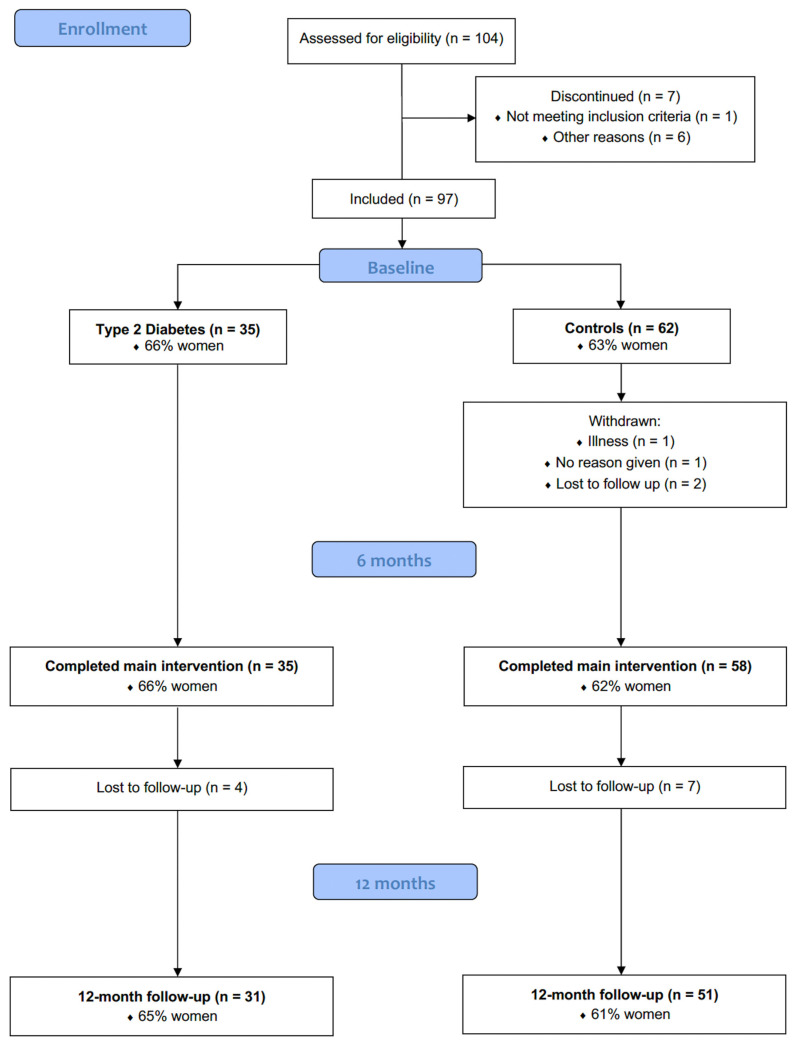
Flowchart of the study with a 6-month intervention and a 12-month follow-up.

**Figure 2 ijms-25-09731-f002:**
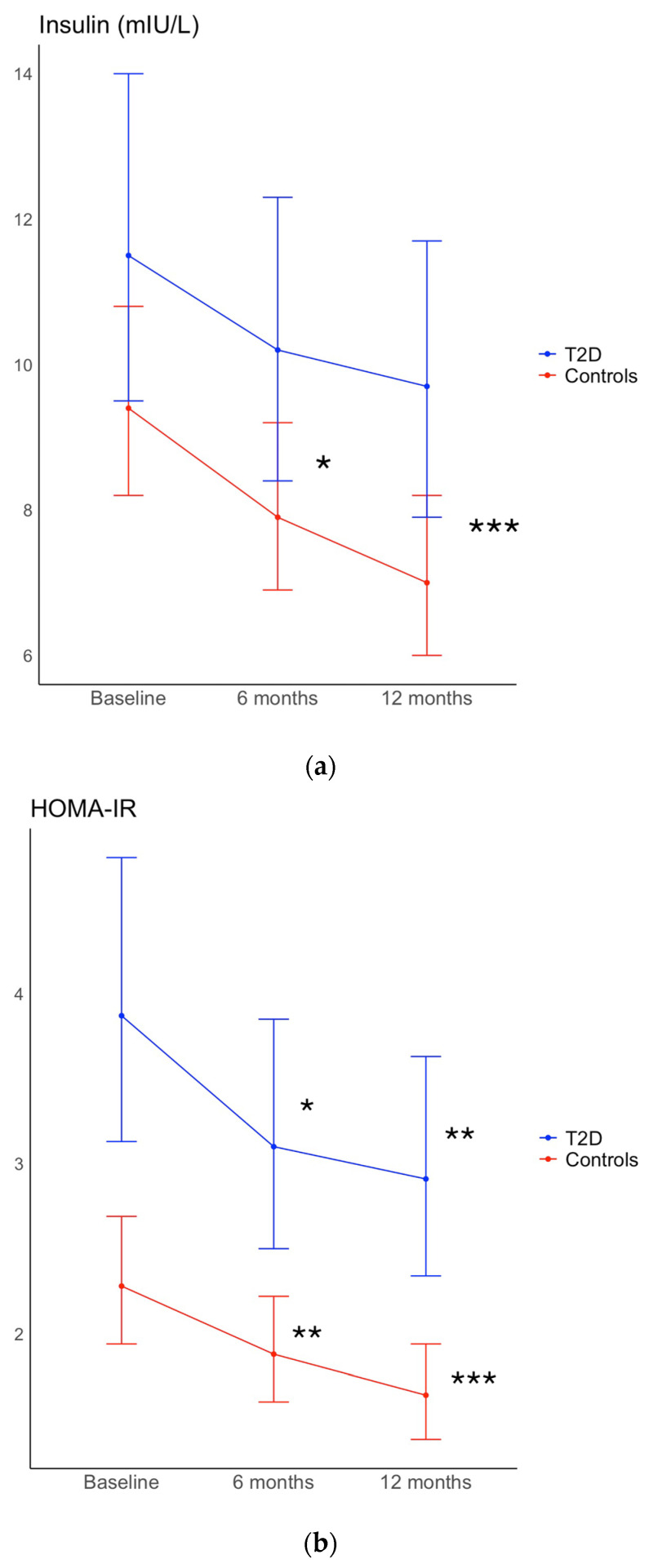
(**a**) Fasting insulin levels and (**b**) HOMA-IR values at baseline, 6 month intervention, and 12-month follow-up for participants with type 2 diabetes (T2D group) and controls, respectively. Within-group differences are represented by * = *p* < 0.05, ** = *p* < 0.01, and *** = *p* < 0.001.

**Table 1 ijms-25-09731-t001:** Baseline characteristics of the 97 subjects with and without T2D.

	T2D	Controls	*p*-Value
No. (% women)	35 (66%)	62 (63%)	
Age (year)	66.63 ± 8.11	57.98 ± 11.76	**<0.001**
Weight (kg)	82.37 ± 15.26	86.35 ± 15.61	0.23
BMI (kg/m^2^)	28.55 ± 3.46	28.85 ± 4.04	0.71
Waist (cm)	103.83 ± 10.50	100.87 ± 10.83	0.19
Body fat (%)	34.78 ± 7.28	34.50 ± 7.97	0.87
Trunk fat (%)	33.73 ± 6.76	34.06 ± 6.94	0.82
Glucose (mmol/L)	7.35 (6.45–8.71)	5.38 (5.04–5.86)	**<0.001**
Insulin (mIU/L)	10.53 (7.88–13.49)	9.64 (6.65–12.55)	0.053
C-peptide (nmol/L)	1.10 (0.90–1.30)	0.82 (0.64–1.00)	**<0.001**
IGF-SD	0.18 ± 0.95	0.05 ± 1.26	0.38
IGFBP-1 (µg/L)	31.14 ± 18.78	32.19 ± 18.34	0.10
HOMA-IR	3.79 (2.65–4.69)	2.28 (1.59–3.33)	**<0.001**
HbA1c	47.0 (43.5–53.0)	35.0 (33.0–38.0)	**<0.001**

Values are presented as mean ± SD or as median and interquartile range (25th–75th). Significant results are presented in bold. BMI = body mass index; C-peptide = connecting peptide; Fat (%) = total body fat (%); HbA1c = glycated hemoglobin; HOMA-IR = homeostatic model assessment for insulin resistance; IGFBP-1 = insulin-like growth factor-binding protein; IGF-SD = age-adjusted insulin-like growth factor; Trunk (%) = trunk fat (%). Glucose, insulin, C-peptide, IGF-SD, IGFBP-1, HOMA, and HbA1c values are presented as fasting values, and statistics are based upon age-adjusted analyses.

**Table 2 ijms-25-09731-t002:** Anthropometric and metabolic variables before and 6 months after intervention in the T2D group and controls.

	T2D (66% Females)			Controls (63% Females)				
	Baseline (n = 35)	6 Months (n = 35)	*p*-Value	Baseline (n = 62)	6 Months (n = 58)	*p*-Value	Difference Diabetics-Controls	*p*-Value
Weight (kg)	84.9 (79.8–89.9)	79.6 (74.5–84.7)	**<0.001**	84.6 (80.8–88.4)	80.4 (76.6–84.3)	**<0.001**	1.05 (−0.26–2.37)	0.12
BMI (kg/m^2^)	29.2 (27.9–30.5)	27.4 (26.1–28.7)	**<0.001**	28.4 (27.5–29.4)	27.0 (26.0–27.9)	**<0.001**	0.39 (−0.05–0.83)	0.08
Waist (cm)	104 (100–108)	99 (95–103)	**<0.001**	101 (98–104)	97 (94–100)	**<0.001**	1.7 (0.25–3.07)	**0.022**
Body fat (%)	35.3 (32.5–38.0)	32.7 (30.0–35.5)	**<0.001**	34.2 (32.1–36.2)	32.2 (30.2–34.3)	**<0.001**	0.60 (−0.34–1.53)	0.21
Trunk fat (%)	34.3 (31.7–36.9)	31.6 (29.1–34.2)	**<0.001**	33.7 (31.8–35.6)	31.4 (29.5–33.3)	**<0.001**	0.44 (−1.20–2.07)	0.60
Glucose^L^ (mmol/L)	7.54 (7.24–7.92)	6.89 (6.55–7.17)	**<0.001**	5.47 (5.26–5.70)	5.31 (5.16–5.47)	0.15	−0.93 (−0.98–−0.89)	**0.004**
Insulin^L^ (mIU/L)	11.5 (9.5–14.0)	10.2 (8.4–12.3)	0.25	9.4 (8.2–10.8)	7.9 (6.9–9.2)	**0.02**	−1.0 (−1.3–−0.9)	0.66
C-peptide^L^ (nmol/L)	1.10 (0.97–1.24)	0.99 (0.87–1.12)	**0.03**	0.83 (0.76–0.91)	0.77 (0.70–0.84)	**0.04**	−0.98 (−1.09–0.88)	0.65
IGF-SD	0.01 (−0.36–0.38)	0.15 (−0.22–0.52)	0.30	0.21 (−0.07–0.50)	−0.05 (−0.34–0.23)	**0.001**	−0.41 (−0.64–−0.17)	**<0.001**
IGFBP-1 (µg/L)	28.0 (22.1–34.0)	34.4 (28.5–40.4)	**0.04**	34.3 (29.9–38.8)	35.0 (30.5–39.5)	0.94	−5.7 (−12.2–0.7)	0.08
HOMA-IR^L^	3.88 (3.13–4.80)	3.10 (2.50–3.85)	**0.023**	2.28 (1.94–2.69)	1.88 (1.60–2.22)	**0.0095**	−0.97 (−1.20–−0.79)	0.79
HbA1c^L^	47.9 (45.6–50.4)	45.2 (42.9–47.5)	**0.004**	36.2 (34.8–37.3)	36.6 (35.1–38.1)	0.67	−0.93 (−0.97–−0.88)	**0.002**

Values are presented as mean (95% CI). Significant results are presented in bold. BMI = body mass index; C-peptide = connecting peptide; Fat (%) = total body fat (%); HbA1c = glycated hemoglobin; HOMA-IR = homeostatic model assessment for insulin resistance; IGF-SD = age-adjusted insulin-like growth factor; Trunk (%) = trunk fat (%). ^L^ denotes log-transformed *p*-values and antilog values. Glucose, insulin, C-peptide, IGF-SD, IGFBP-1, HOMA, and HbA1c values are presented as fasting values. Statistics are performed by Mixed Model ANOVA and based on ITT and age-adjusted analyses.

**Table 3 ijms-25-09731-t003:** Anthropometric and metabolic variables before and 12 months after intervention in the T2D group and controls.

	T2D (66% Females)			Controls (62% Females)		
	Baseline (n = 35)	12 Months (n = 31)	*p*-Value	Baseline (n = 62)	12 Months (n = 51)	*p*-Value
Weight (kg)	84.9 (79.8–89.9)	81.0 (75.9–86.1)	**<0.001**	84.6 (80.8–88.4)	81.9 (78.1–85.8)	**<0.001**
BMI (kg/m^2^)	29.2 (27.9–30.5)	27.8 (26.5–29.1)	**<0.001**	28.4 (27.5–29.4)	27.5 (26.5–28.5)	**<0.001**
Waist (cm)	104 (100–108)	100 (96–104)	**<0.001**	101 (98–104)	97 (94–100)	**<0.001**
Body fat (%)	35.3 (32.5–38.0)	33.8 (31.0–36.6)	**0.001**	34.2 (32.1–36.2)	32.9 (30.9–35.0)	**<0.001**
Trunk fat (%)	34.3 (31.7–36.9)	32.2 (29.6–34.8)	**0.006**	33.7 (31.8–35.6)	31.7 (29.8–33.7)	**0.001**
Glucose^L^ (mmol/L)	7.54 (7.24–7.92)	6.88 (6.55–7.24)	**<0.001**	5.47 (5.26–5.70)	5.26 (5.05–5.47)	**0.049**
Insulin^L^ (mIU/L)	11.5 (9.5–14.0)	9.7 (7.9–11.7)	0.06	9.4 (8.2–10.8)	7.0 (6.0–8.2)	**<0.001**
C-peptide^L^ (nmol/L)	1.10 (0.97–1.24)	1.10 (0.97–1.25)	0.99	0.83 (0.76–0.91))	0.80 (0.73–0.88)	0.59
IGF-SD	0.01 (−0.36–0.38)	0.25 (−0.12–0.63)	**0.03**	0.21 (−0.07–0.50)	0.07 (−0.22–0.36)	0.17
IGFBP-1 (µg/L)	28.0 (22.1–34.0)	29.4 (23.2–35.5)	0.87	34.3 (29.9–38.8)	38.0 (33.3–42.7)	0.20
HOMA-IR^L^	3.88 (3.13–4.80)	2.91 (2.34–3.63)	**0.004**	2.28 (1.94–2.69)	1.64 (1.38–1.94)	**<0.001**
HbA1c^L^	47.9 (45.6–50.4)	46.1 (43.8–48.4)	0.14	36.2 (34.8–37.3)	36.6 (35.1–38.1)	0.66

Values are presented as mean (95% CI). Significant results are presented in bold. ANOVA = analysis of variance; BMI = body mass index; C-peptide = connecting peptide; Fat (%) = total body fat (%); HbA1c = glycated hemoglobin; HOMA-IR = homeostatic model assessment for insulin resistance; IGF-SD = age-adjusted insulin-like growth factor; Trunk (%) = trunk fat (%). ^L^ denotes log-transformed *p*-values and antilog values. Glucose, insulin, C-peptide, IGF-SD, IGFBP-1, HOMA, and HbA1c values are presented as fasting values. Statistics are performed by Mixed Model ANOVA and based on ITT and age-adjusted analyses.

## Data Availability

The raw data supporting the conclusions of this article will be made available by the authors on reasonable request.

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
