# Peer review of "The 5:2 Diet Affects Markers of Insulin Secretion and Sensitivity in Subjects with and without Type 2 Diabetes—A Non-Randomized Controlled Trial"

_ijms, 2024, doi:10.3390/ijms25179731_

Round 1

Reviewer 1 Report

Comments and Suggestions for Authors

Summary: In the current study, the authors linked an intermittent calorie restriction (5:2 diet method) with diabetes-related factors such as obesity, waist size, weight, etc. The study involves one phase of 6-month interventions and a second phase of 12-month follow-ups. The subjects exposed to interventions were assessed for diabetes markers and compared for any beneficial effects. The authors observed the beneficial effects of intermittent calorie restrictions on glycemic control based on the evaluation of signature molecules of diabetes. The manuscript is well-written and overall good; This reviewer has a few comments that must be addressed.

Comment (1)

In section 2.2 namely six months interventions (lines 108-109), the authors reported that “Fasting glucose decreased significantly only in the T2D group (P < 0.001), while insulin levels decreased significantly only in the control group (P = 0.02), and C-peptide decreased in both groups (T2D: P = 0.03, Controls: P = 0.04) (Table 2)”. Based on these observations of lowering glucose levels in T2D and insulin levels in healthy subjects, how do the authors discuss/conclude this observation? Is lowering glucose levels in T2D exhibited by higher insulin release or fasting making both glucose and insulin levels drop? Similarly, reduced insulin levels observed in control confers lower glucose levels in fasting subjects. What about insulin sensitivity?

Comment (2)

Result Section (line 142-143), the authors further observed that “Fasting glucose were furthermore still significantly reduced in both groups and insulin in the Control group, but C-peptide was no longer reduced in the two groups at 12 months compared to baseline (Table 3, Figure 2).” This reviewer suggests to integrate this with comment (3).

Comment (3)

There is a connection between calorie restriction with modifiable factors (diabetes determinants); however, I noted that after 6 and 12 months the connection between calorie restriction and modifiable factors was lost but the effect on improving glucose homeostasis persisted. The authors need to explain this with a possible conclusion in the discussion part.

Comment (4)

Result Section (lines 131-136); the suggestion is to verify the information of study subjects in contrast to the information depicted in Figure 1. The authors mention that three in the T2D and five in the control were lost in the follow-up. Figure 1 shows 4 lost in T2D and  7 in the control were lost in the follow-up.

Comment (5)

In the method section of the subject (lines 269-275), the authors include diabetic patients on treatment with different classes of anti-diabetic medications; however, these comparators are overlooked in the main body of results and discussion.

Comment (6)

Did the authors search the literature regarding information on men? Is it possible to have any gender differences? This might be discussed in the “Discussion” part.

Comment (7)

The discussion section needs a little more work to integrate calorie restriction, modifiable factors, and diabetes markers in an articulated manner to convince the readers.

Comment (8)

In Figure 2, the significance can be shown with asterisks at data points.

Comment (9)

This might be great if, along with the insulin tolerance test, an oral glucose tolerance test is conducted especially after 6 months (no restriction period) to draw a complete picture of insulin resistance and secretion.

Minor comments:

Comment- Ensure the numbering of the subsections, as I noticed 2.2 is duplicated in the result section. Similarly, in material and methods, the subsections are NOT arranged appropriately.

Comment- An improved high-resolution Figure is needed for Fig1 (flow-chart).

Comment- References section; References are OK but this reviewer recommends adding more recent literature since there are only 4 studies shown after 2020 and some between 2015 and 2020. I believe that there are more recent studies that might support some of these findings.

Author Response

We thank the reviewers for the comments and suggestions, and we have now to the best of our knowledge carefully considered and addressed all suggestions and comments. Changes are highlighted in yellow in the manuscript. 

We think that the manuscript is clearly improved, and we now hope that the manuscript can be accepted for publication in the International Journal of Molecular Sciences.

Point-by-point response to the reviewer comments:

Comment 1:

In section 2.2 namely six months interventions (lines 108-109), the authors reported that “Fasting glucose decreased significantly only in the T2D group (P < 0.001), while insulin levels decreased significantly only in the control group (P = 0.02), and C-peptide decreased in both groups (T2D: P = 0.03, Controls: P = 0.04) (Table 2)”. Based on these observations of lowering glucose levels in T2D and insulin levels in healthy subjects, how do the authors discuss/conclude this observation? Is lowering glucose levels in T2D exhibited by higher insulin release or fasting making both glucose and insulin levels drop? Similarly, reduced insulin levels observed in control confers lower glucose levels in fasting subjects. What about insulin sensitivity?

Response 1: Thank you for the comment. The C-peptide and glucose levels were higher at baseline in those with T2D suggesting more pronounced insulin resistance compared to the controls. However, in both groups the C-peptide levels decreased significantly after six months of intermittent fasting, and fasting glucose was not significantly decreased (P = 0.15) in the controls but significantly decreased in T2D (p<0.001). Thus, the insulin sensitivity was improved in both groups, which also was shown by decreased HOMA-IR in the groups. Thus, after six months intervention the same insulin secretion was sufficient to reduce fasting glucose in T2D, and the reduced insulin secretion was enough to keep fasting glucose within normal range (5.3 mmol/l, CI 5.16-5.47) in the controls. C-peptide and insulin are secreted to v. porta in equimolar amounts from the beta cells when stimulated by high glucose. C-peptide escapes the hepatic extraction, while circulating insulin, which we measure, is the result of both insulin secretion and the insulin degradation in the liver. The blunted decrease in insulin but not in C-peptide suggest that the reduced hepatic insulin extraction was not improved by the 5:2 diet in T2D in contrast to that in the controls. This is now elaborated on in the Discussion.

Comment 2:

Result Section (line 142-143), the authors further observed that “Fasting glucose were furthermore still significantly reduced in both groups and insulin in the Control group, but C-peptide was no longer reduced in the two groups at 12 months compared to baseline (Table 3, Figure 2).” This reviewer suggests to integrate this with comment (3).

Response 2: See below #3.

Comment 3:

There is a connection between calorie restriction with modifiable factors (diabetes determinants); however, I noted that after 6 and 12 months the connection between calorie restriction and modifiable factors was lost but the effect on improving glucose homeostasis persisted. The authors need to explain this with a possible conclusion in the discussion part.

Response to #2 and #3: Thank you for the comment. Calorie restriction leads to weight loss and decreased body fat. The majority of the participants reported that they had followed a calorie restriction after the intervention until the 12-month visit. The weight loss was still significant in both groups, and so was the decrease in waist circumference, body and trunk fat percent, which suggest improved hepatic insulin sensitivity at 12 months. The insulin resistance measured by HOMA-IR was still significantly lower compared to baseline with decreased fasting glucose and insulin levels. The C-peptide levels, however, had increased to the baseline levels in both groups. Together this suggests that the hepatic insulin extraction had increased, which is further supported by a blunted increase in IGFBP-1. An improved insulin extraction suggests improved hepatic insulin sensitivity, which explains the improved fasting glucose levels. This is now added to the Discussion.

Comment 4:

Result Section (lines 131-136); the suggestion is to verify the information of study subjects in contrast to the information depicted in Figure 1. The authors mention that three in the T2D and five in the control were lost in the follow-up. Figure 1 shows 4 lost in T2D and 7 in the control were lost in the follow-up.

Response 4: We apologize for the mistake. It should be four participants (three women) lost to follow-up in the T2D group and seven (five women) in the control group (Figure 1). It is now corrected.

Comment 5:

In the method section of the subject (lines 269-275), the authors include diabetic patients on treatment with different classes of anti-diabetic medications; however, these comparators are overlooked in the main body of results and discussion.

Response 5: Three subjects with diabetes had been treated with GLP1 receptor agonist (liraglutide) once a day and two with SGLT2i for more than six months prior to the intervention and their results did not differ from the mean values. The controls were not treated with any appetite regulating drug.

Comment 6:

Did the authors search the literature regarding information on men? Is it possible to have any gender differences? This might be discussed in the “Discussion” part.

Response 6: There are studies on men and women but, to our knowledge, any different response to intermittent fasting has not been reported. Furthermore, there were no gender differences in our study (data not shown). However, we cannot exclude that the lack of significant differences could at least partly be due to small subgroups.

Comment 7:

The discussion section needs a little more work to integrate calorie restriction, modifiable factors, and diabetes markers in an articulated manner to convince the readers.

Response 7: The calorie restriction during the 5:2 intervention led to reduced body weight, body fat and visceral fat (trunk fat) associated with reduced insulin secretion and improved insulin sensitivity in overweight/obese subjects both with T2D and without abnormal glucose control. This explains the reduced fasting glucose and HbA1c in T2D, which is now added to the Discussion. We have included HbA1c as an important diabetes marker. HbA1c is a marker of both postprandial and fasting glucose over several weeks, while fasting glucose is a marker of hepatic gluconeogenesis regulated by hepatic insulin sensitivity. HbA1c was reduced after six-month-intervention in the T2D subjects but not in the controls with normal HbA1c levels. However, at 12 months, HbA1c had increased and was not any longer significantly decreased compared to baseline in T2D, please see the Results and Discussion sections.

Comment 8:

In Figure 2, the significance can be shown with asterisks at data points.

Response 8: The asterisks are now added in figure 2A and B.

Comment 9:

This might be great if, along with the insulin tolerance test, an oral glucose tolerance test is conducted especially after 6 months (no restriction period) to draw a complete picture of insulin resistance and secretion.

Response 9:  Thank you for the comment. We did not perform any insulin tolerance test or an OGT-test, which is a limitation, now added to the Discussion. However, we have used IGFBP-1 as a marker for hepatic and whole-body insulin sensitivity. IGFBP-1 is regulated at the transcriptional level by insulin, inhibiting its production and secretion from the liver. Furthermore, IGF-I is used as a marker for insulin secretion. The production of hepatic IGF-I is regulated by growth hormone, with insulin and amino acids as important co-factors. This is considered a limitation of the study and is now addressed in the Discussion.

Minor comments:

Comment- Ensure the numbering of the subsections, as I noticed 2.2 is duplicated in the result section. Similarly, in material and methods, the subsections are NOT arranged appropriately.

Response:  This is now corrected. 

Comment- An improved high-resolution Figure is needed for Fig1 (flow-chart).

Response: This is now corrected; the image now has 600 dpi.

Comment- References section; References are OK but this reviewer recommends adding more recent literature since there are only 4 studies shown after 2020 and some between 2015 and 2020. I believe that there are more recent studies that might support some of these findings.

Response: We have already included a review article from 2022. After this date we have found a narrative review which we now have added as well as a systematic review and meta-analysis from 2021. 

Reviewer 2 Report

Comments and Suggestions for Authors

Calorie restriction (CR) is the only scientifically validated method of extending lifespan and healthspan across a wide range of organisms, from yeast all the way to humans. Chronic CR typically consists of restricting daily calorie consumption by 300-600 kCal, which is impractical for most humans given sociocultural norms around eating and drinking. Intermittent CR is a way around this issue, with 5:2 intermittent CR consisting of ad libitum feeding without eating to excess over 5 days and restricted calorie intake on 2 days. The following study examined 5:2 CR on insulin or insulin-related markers in people with vs. without type 2 diabetes. Overall, this is a useful study that can certainly add to a sparse literature. I do have some key concerns or suggestions, as mentioned below.

The introduction is good and offers a typical perspective on why alternate forms of CR relative to chronic CR is done. It also explains why HOMA, IGF, and related hormones are analyzed in the data.

For methods, 6 months of intermittent CR should be more than sufficient to see an effect of interest. The 12 month washout period is useful for seeing which effects persist and which do not with normal nutrition or participant-directed CR. Exclusion criteria are few and this makes sense, because recruiting participants for a long-term CR study is painstaking and takes a very long time (as briefly suggested under section 4.1). The dietary intervention sounds right, though 2 non-consecutive days could affect metabolic measures. I hate to ask the following, because it always comes up, but did the authors use beta-hydroxybutyrate or some other index of fasting to biologically determine if participants were in or out of compliance during the study? The HOMA measure of 2.9 indicating insulin resistance seems high to me, as the original HOMA paper suggested something like 1.8 being the cutoff. However, that work is nearly 40 years old and HOMA2 while more exact is still not terribly specific. The stats section for non-parametric and parametric analyses looks fine.

For results, the type 2 diabetes and control groups significantly differ by age. Did the authors covary for age in their statistical analyses? In the stats section there are no covariates listed, but under Table 2 there is the suggestion that analyses were done with or without age adjustment. In any case, if age was not adjusted, this presents a serious confound and all results must be redone with this covariate. This is particularly because trunk adiposity and adipokine release increases substantially with age and in particular after menopause or andropause.

The 12 month follow-up is also interesting because there’s a trifurcation of adults who either continue CR, sometimes do it, or do not do it at all anymore. For sensitivity or follow-up analyses, it would be interesting to compare anthropometric, adipokine, and other relevant biomarkers against the three latent groups at the end of the washout period.

The discussion section summarizes results well and presents a useful contrast with the handful of longitudinal CR studies done in the past. I have no suggestions here.

Author Response

Responses to the reviewers and the editor

We thank the reviewers for the comments and suggestions, and we have now to the best of our knowledge carefully considered and addressed all suggestions and comments. Changes are highlighted in yellow in the manuscript. 

We think that the manuscript is clearly improved, and we now hope that the manuscript can be accepted for publication in the International Journal of Molecular Sciences.

Point-by-point response to the reviewer comments:

Calorie restriction (CR) is the only scientifically validated method of extending lifespan and health span across a wide range of organisms, from yeast all the way to humans. Chronic CR typically consists of restricting daily calorie consumption by 300-600 kcal, which is impractical for most humans given sociocultural norms around eating and drinking. Intermittent CR is a way around this issue, with 5:2 intermittent CR consisting of ad libitum feeding without eating to excess over 5 days and restricted calorie intake on 2 days. The following study examined 5:2 CR on insulin or insulin-related markers in people with vs. without type 2 diabetes. Overall, this is a useful study that can certainly add to a sparse literature. I do have some key concerns or suggestions, as mentioned below.

The introduction is good and offers a typical perspective on why alternate forms of CR relative to chronic CR is done. It also explains why HOMA, IGF, and related hormones are analyzed in the data.

For methods, 6 months of intermittent CR should be more than sufficient to see an effect of interest. The 12 months washout period is useful for seeing which effects persist and which do not with normal nutrition or participant-directed CR. Exclusion criteria are few and this makes sense, because recruiting participants for a long-term CR study is painstaking and takes a very long time (as briefly suggested under section 4.1). The dietary intervention sounds right, though 2 non-consecutive days could affect metabolic measures.

Response: Thank you for the positive comments on our study.

I hate to ask the following, because it always comes up, but did the authors use beta-hydroxybutyrate or some other index of fasting to biologically determine if participants were in or out of compliance during the study?

Response: Unfortunately, we did not measure beta hydroxybutyrate or ketones during the fasting days, which could have supported the fasting. We have only the written report of what menu they had chosen during the fasting days.  This is considered a limitation of the study and is now addressed in the Discussion.

The HOMA measure of 2.9 indicating insulin resistance seems high to me, as the original HOMA paper suggested something like 1.8 being the cutoff. However, that work is nearly 40 years old and HOMA2 while more exact is still not terribly specific.

Response: HOMA-IR was improved in T2D, but not normalized at the 12 months visit, while HOMA-IR was further improved to reach more normal values at 12 months in the controls. maybe due to more weight loss (4%) at 12 months in the controls compared to those with T2D (2%). This is now addressed in the Discussion.

The stats section for non-parametric and parametric analyses looks fine.

For results, the type 2 diabetes and control groups significantly differ by age. Did the authors covary for age in their statistical analyses? In the stats section there are no covariates listed, but under Table 2 there is the suggestion that analyses were done with or without age adjustment. In any case, if age was not adjusted, this presents a serious confound and all results must be redone with this covariate. This is particularly because trunk adiposity and adipokine release increases substantially with age and in particular after menopause or andropause.

Response: The results in table 2 and 3 are adjusted for age as a covariate in the ANOVA mixed model analyses. This is described in the Statistical section. Even though there was a difference in age between the two groups, they were composed by primarily postmenopausal women.

The 12 months follow-up is also interesting because there’s a trifurcation of adults who either continue CR, sometimes do it, or do not do it at all anymore. For sensitivity or follow-up analyses, it would be interesting to compare anthropometric, adipokine, and other relevant biomarkers against the three latent groups at the end of the washout period.

Response: We fully agree, however this is not the scope of the present study, but will be further analyzed in a future study.

The discussion section summarizes results well and presents a useful contrast with the handful of longitudinal CR studies done in the past. I have no suggestions here.

Response: Thank you very much for all valuable comments. 

Round 2

Reviewer 2 Report

Comments and Suggestions for Authors

The authors have done a fine job addressing my comments. I have nothing further to add.